# CLIP-MUSED: CLIP-Guided Multi-Subject Visual Neural Information Semantic Decoding

Qiongyi Zhou[1,2], Changde Du[1], Shengpei Wang[1] and Huiguang He[1,2,*]

[1]Laboratory of Brain Atlas and Brain-Inspired Intelligence, State Key Laboratory of Multimodal
Artificial Intelligence Systems, Institute of Automation, Chinese Academy of Sciences
[2]University of Chinese Academy of Sciences
[*]Corresponding Author
`zhouqiongyi@hotmail.com`, `{changde.du, wangshengpei2014,`
`huiguang.he}@ia.ac.cn`

## Abstract

The study of decoding visual neural information faces challenges in generalizing single-subject decoding models to multiple subjects, due to individual differences. Moreover, the limited availability of data from a single subject has a constraining impact on model performance. Although prior multi-subject decoding methods have made significant progress, they still suffer from several limitations, including difficulty in extracting global neural response features, linear scaling of model parameters with the number of subjects, and inadequate characterization of the relationship between neural responses of different subjects to various stimuli. To overcome these limitations, we propose a **CLIP**-guided **M**ulti-s**U**bject visual neural information **SE**mantic **D**ecoding (CLIP-MUSED) method. Our method consists of a Transformer-based feature extractor to effectively model global neural representations. It also incorporates learnable subject-specific tokens that facilitates the aggregation of multi-subject data without a linear increase of parameters. Additionally, we employ representational similarity analysis (RSA) to guide token representation learning based on the topological relationship of visual stimuli in the representation space of CLIP, enabling full characterization of the relationship between neural responses of different subjects under different stimuli. Finally, token representations are used for multi-subject semantic decoding. Our proposed method outperforms single-subject decoding methods and achieves state-of-the-art performance among the existing multi-subject methods on two fMRI datasets. Visualization results provide insights into the effectiveness of our proposed method. Code is available at https://github.com/CLIP-MUSED/CLIP-MUSED.

## 1 Introduction

In recent years, researchers have made significant progress in visual neural decoding tasks, allowing for the deciphering of semantic information from brain activities in response to visual stimuli (Akamatsu et al., 2020; Li et al., 2022; Bagchi & Bathula, 2022). However, due to individual differences, most decoding models are trained separately on each subject. Single-subject models are susceptible to overfitting due to the limited data available for each subject as a result of constraints in data acquisition. Furthermore, single-subject models exhibit weak generalization performance on new subjects. In contrast, multi-subject decoding methods can aggregate data from multiple subjects, mitigate overfitting issues, and achieve superior performance across different subjects. Thus, it is of great value to investigate multi-subject neural information decoding methods.

Individual differences manifest in both the anatomical structure and functional topography of the brain (Haxby et al., 2020). While image registration techniques can eliminate anatomical differences, differences in functional topography persist, including variations in the size, shape, and location of

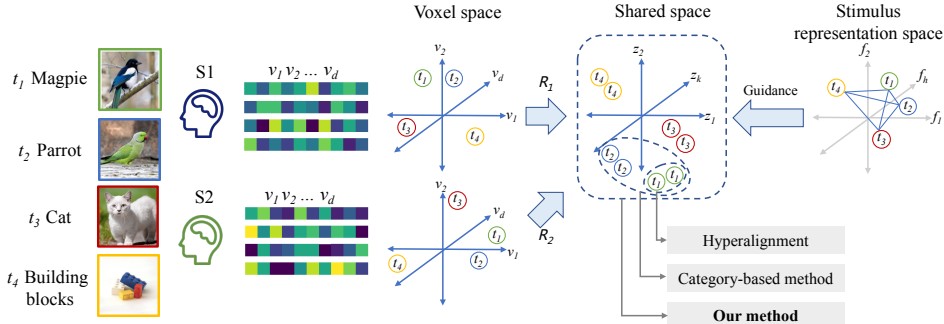

Figure 1: Diagram of the different multi-subject functional alignment methods.

functional areas on the cortical surface. Therefore, aligning the functional topology of different subjects poses a critical challenge in multi-subject decoding research.

Most multi-subject decoding methods are based on hyperalignment (Haxby et al., 2020; Chen et al., 2015), a classic method for functional alignment. As shown in Fig. 1, subjects $S_1$ and $S_2$ view four stimuli ($t_1 \sim t_4$), which are represented as high-dimensional vectors in the voxel space of each subject. Hyperalignment transforms neural responses from the voxel space of each subject to a shared space through the learning of a mapping function $R_j$. It aligns the functional topology structure of different subjects by bringing together the neural representations of different subjects under the same stimulus in the shared space, as illustrated by the same-colored balls in Fig. 1. Nonetheless, hyperalignment cannot handle the common scenario where different subjects view different stimuli during data acquisition. To tackle this challenge, Li et al. (2020) proposed a category-based functional alignment approach. As shown in Fig. 1, this method pulls together the neural representations of different subjects in the shared space that relate to the same category of stimuli (two bird images $t_1$ and $t_2$).

However, these multi-subject decoding methods still have three limitations:

1. The mapping function $R_i$ has restricted expressive power. Linear transformation discussed by Yousefnezhad & Zhang (2017) and MLP composed of stacked linear layers used by Yousefnezhad & Zhang (2017) are unsuitable for high-dimensional voxel responses. To overcome this limitation, Chen et al. (2016) proposed a CNN-based hyperalignment algorithm. However, CNNs face challenge in capturing global features that reflect the long-range functional connectivities between brain regions.

2. Current studies require learning a distinct mapping function $R_i$ for each subject. As the number of subjects increases, the number of model parameters will also increase linearly, leading to a considerable increase in computational complexity.

3. Existing methods have not fully characterized the relationship between the neural responses of different subjects under similar stimuli. However, research has demonstrated that different subjects exhibit similar neural responses when presented with semantically analogous visual stimuli (Huth et al., 2012; Connolly et al., 2012; Carlson et al., 2014; Zhang et al., 2020). For instance, in Fig. 1, $t_1$ and $t_2$ are birds, while $t_3$ (cat) and $t_4$ (building blocks) are not. Additionally, birds and cats both belong to the animal category, whereas building blocks are inanimate objects. Therefore, in the shared space, the representation of $t_3$ should be more proximate to the representations of $t_1$ and $t_2$ than that of $t_4$.

To address the aforementioned issues, we propose the following solutions:

1. Transformers possess the ability to capture long-range dependencies. Therefore, we design a Transformer-based network to learn the mapping function and model the relationships between different brain regions.

2. Transformers can flexibly add tokens that learn specific knowledge. Prior research has introduced diverse inductive biases into Transformers through extra tokens (Touvron et al., 2021; Naseer et al., 2021). Inspired by these studies, we introduce subject-specific tokens into the Transformer model in this paper. These learnable tokens encode individual differences, allowing other parameters to be shared across subjects.

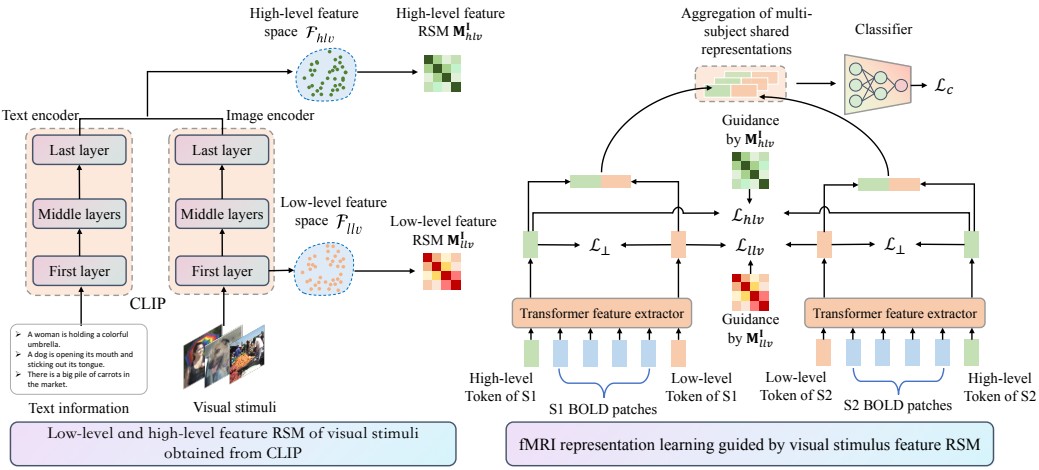

Figure 2: The framework of the proposed method. Left: Low-level and high-level feature RSM of visual stimuli are obtained from CLIP at first. Right: The Transformer-based encoder extracts multi-subject shared neural representations guided by the visual stimulus feature RSM.

3. Previous studies (Zhou et al., 2022) have shown that CLIP (Radford et al., 2021) has stronger explanatory power for neural responses in the ventral visual pathway than single-modal DNNs, indicating high consistency between CLIP and cortical representations of visual stimuli. Thus, the topological relationships of visual stimuli in the CLIP representation space can serve as prior knowledge to guide neural representation learning in the shared space.

In summary, we propose a **CLIP**-guided **M**ulti-s**U**bject visual neural information **SE**mantic **D**ecoding method (CLIP-MUSED), as illustrated in Fig. 2. The proposed method leverages a Transformer-based fMRI feature extractor to map the neural responses of each subject from the original voxel space to the shared space. We further divide individual differences into two categories: differences in the processing patterns of low-level features (such as shape and color) and high-level features (such as semantic categories) of visual stimuli. To encode these two types of differences, we incorporate a low-level token and a high-level token for each subject into the Transformer architecture. The proposed method uses the topological relationships of visual stimuli in the shallower and deeper layers of CLIP to guide the representation learning of low-level and high-level tokens, respectively, through representational similarity analysis (RSA). To ensure that the low-level and high-level token representations of the same subject encode as much different information as possible, we impose an orthogonal constraint on two representations inspired by the previous study (Niu et al., 2019). Given that both low-level and high-level features play critical roles in semantic classification, we concatenate the low-level and high-level token representations of each subject for semantic classification.

The contributions are summarized as follows.

1. We propose a Transformer-based fMRI feature extractor that can efficiently extract global features of neural responses.

2. CLIP-MUSED learns low-level and high-level tokens for each subject and shares the other parameters across subjects. The method can be applied to multiple subjects without linear increase of parameters.

3. With the help of RSA, CLIP-MUSED utilizes the topological relationships of visual stimuli in the CLIP representation space to fully characterize the relationship between neural representations under different stimuli for different subjects.

4. The experimental results on two fMRI datasets demonstrate that CLIP-MUSED surpasses single-subject decoding methods by aggregating more training data and reducing individual differences. Our method also achieves state-of-the-art performance among the existing multi-subject methods.

## 2 Methodology

### 2.1 Overview

Given a neural dataset with neural activities of $N$ subjects under visual stimuli, let $\mathcal{X}^{(n)}$ denote the voxel space. $\mathbf{X}^{(n)} \in \mathbb{R}^{n_i \times d_i}$ is the neural responses of the $n$-th subject, where $n_i$ is the number of image stimuli and $d_i$ is the number of voxels. Let $\mathcal{I}$ and $\mathcal{Y}$ denote the pixel space and the label space of the image stimuli, respectively. Our goal is to aggregate the neural responses of the $N$ subjects to train a classifier $C : \mathcal{X}^{(1)} \times \cdots \times \mathcal{X}^{(N)} \rightarrow \mathcal{Y}$. To achieve this goal, we propose CLIP-MUSED that first maps the image stimuli from $\mathcal{I}$ to the representation space $\mathcal{F}$ of CLIP. Then, a Transformer feature extractor is used to map the neural responses from $\mathcal{X}^{(n)}$ to a shared space $\mathcal{Z}$. Finally, RSA is employed to guide the representation learning of $\mathcal{Z}$ using the topological relationships of the visual stimuli in $\mathcal{F}$.

### 2.2 CLIP-based feature extraction of visual stimuli

In a previous study (Zhou et al., 2022), it was shown that CLIP (Radford et al., 2021) outperforms various single-modal DNNs in explaining cortical activity, and that there is a hierarchical correspondence between CLIP and the human ventral visual pathway. Based on these findings, we use CLIP to extract features of visual stimuli in our method. However, it is worth noting that other DNN feature spaces could also be used in our approach.

CLIP comprises an image encoder and a text encoder, as shown in Fig. 2. In our method, we input visual stimuli along with corresponding textual information (either textual descriptions or label names) into CLIP to obtain multi-modal features. Due to the hierarchical architecture of CLIP, we use the first-layer features of the image encoder as the low-level feature $\mathbf{f}_{llv}$, while the average of the image and text features from the last layer of CLIP is used as the multi-modal high-level feature $\mathbf{f}_{hlv}$. We compute representation similarity matrices (RSMs), $\mathbf{M}_{llv}^{\mathbf{I}}$ and $\mathbf{M}_{hlv}^{\mathbf{I}}$, to quantify the similarity between $B$ visual stimuli in low-level and high-level feature spaces, where $B$ denotes the batch size in a mini-batch. Specifically, $\mathbf{M}_{llv}^{\mathbf{I}}[i,j]$ and $\mathbf{M}_{hlv}^{\mathbf{I}}[i,j]$ represent the cosine similarity between the $i$th and $j$th images in the feature spaces $\mathcal{F}_{llv}$ and $\mathcal{F}_{hlv}$, respectively.

### 2.3 Transformer-based fMRI feature extraction

The feature extraction process of the conventional Transformer can be formulated as follows:

$$\mathbf{z}_0 = \left[ \mathbf{x}_{class}; \mathbf{x}^1 \mathbf{E}; \cdots ; \mathbf{x}^M \mathbf{E} \right] + \mathbf{E}_{pos}, \tag{1}$$

$$\mathbf{z}'_l = \text{MHSA}\left( \text{LN}\left( \mathbf{z}_{l-1} \right) \right) + \mathbf{z}_{l-1}, \qquad l = 1, 2, \ldots, L \tag{2}$$

$$\mathbf{z}_l = \text{MLP}\left( \text{LN}\left( \mathbf{z}'_l \right) \right) + \mathbf{z}'_l, \qquad l = 1, 2, \ldots, L \tag{3}$$

$$\mathbf{z} = \text{LN}\left( \mathbf{z}_L^0 \right). \tag{4}$$

First, the input data $\mathbf{x} \in \mathbb{R}^d$ is divided into $M$ equally sized patches, resulting in $\mathbf{x} \in \mathbb{R}^{M \times d_{in}}$. As shown in Eq. (1), the patch embeddings of $\mathbf{x}$ are obtained by passing them through a linear embedding layer $\mathbf{E} \in \mathbb{R}^{d_{in} \times d_{out}}$. The resulting patch embeddings are then concatenated with a learnable class Token $\mathbf{x}_{class}$ and fed into an $L$-layer Transformer encoder. To preserve the positional relationships in the original data, positional encoding $\mathbf{E}_{pos} \in \mathbb{R}^{(M+1) \times D_{out}}$ is used. As shown in Eqs. (2) and (3), each layer consists of a multi-head self-attention (MHSA) module with residual connections and a feed-forward (MLP) module. The features are layer-normalized (LN) before being input into the MHSA and MLP modules. As shown in Eq. (4), the model applies layer normalization to the variable $\mathbf{z}_L^0$ and generates the final representation $\mathbf{z}$.

To ensure that our model is applicable to different subjects, we design a Transformer-based fMRI feature extractor with subject-specific tokens, as depicted in Fig. 3. In the input stage, we patchify the blood oxygenation level dependent (BOLD) signals at first. There are two ways to patchify the BOLD signals. The first method involves patchifying the data based on regions of interest (ROIs), while the second method directly patchifies the 3-D BOLD volumes. Fig. 3(a) illustrates the second method. Due to the high dimensionality of the BOLD volumes, directly patchifying them can result in a large number of patches, which leads to high computation cost. To address this problem, we first reduce the dimensionality of the BOLD volumes using a 3D-CNN, and then patchify and

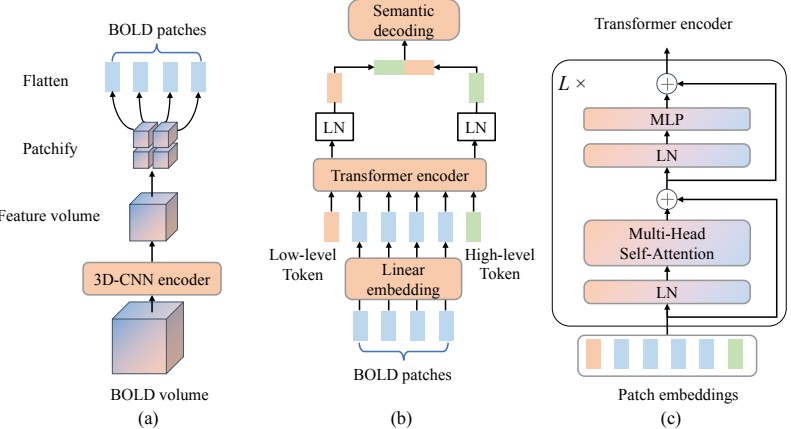

Figure 3: Transformer-based fMRI feature extractor of CLIP-MUSED. (a) Conversion of BOLD signals in Volume format to BOLD patches. (b) Flowchart of the feature extraction process. (c) Network structure of the Transformer encoder.

flatten the feature volumes as BOLD patches. The framework of the Transformer-based feature extractor is shown in Fig. 3(b). Distinct from the conventional Transformer, the model includes learnable subject-specific low-level and high-level tokens $\mathbf{x}_{llv}$ and $\mathbf{x}_{hlv}$. The network structure of the Transformer encoder in Fig. 3(b) is illustrated in Fig. 3(c). Each layer consists of a residual MHSA module and a feed-forward module. The feature extraction process of the Transformer-based multi-subject fMRI feature extractor can be formalized as follows:

$$\mathbf{z}_0 = \left[\mathbf{x}_{llv}; \mathbf{x}_{hlv}; \mathbf{x}^1 \mathbf{E}; \cdots; \mathbf{x}^M \mathbf{E}\right] + \mathbf{E}_{pos}, \tag{5}$$

$$\mathbf{z}'_l = \mathrm{MHSA}\left(\mathrm{LN}\left(\mathbf{z}_{l-1}\right)\right) + \mathbf{z}_{l-1}, \qquad\qquad l = 1, 2, \ldots, L \tag{6}$$

$$\mathbf{z}_l = \mathrm{MLP}\left(\mathrm{LN}\left(\mathbf{z}'_l\right)\right) + \mathbf{z}_{l-1}, \qquad\qquad l = 1, 2, \ldots, L \tag{7}$$

$$\mathbf{z}_{llv} = \mathrm{LN}\left(\mathbf{z}_L^0\right), \tag{8}$$

$$\mathbf{z}_{hlv} = \mathrm{LN}\left(\mathbf{z}_L^1\right). \tag{9}$$

As described in Eq. (6) and Eq. (7), the low-level and high-level tokens interact with different BOLD patches, eventually forming low-level and high-level neural representations denoted as $\mathbf{z}_{llv}$ and $\mathbf{z}_{hlv}$. To extract features from the neural signals $\mathbf{X}^{(n)}$ of the $n$-th subject, CLIP-MUSED calls the low-level token $\mathbf{x}_{llv}^{(n)}$ and high-level token $\mathbf{x}_{hlv}^{(n)}$ of the $n$-th subject. These tokens are then combined with the BOLD patches and fed into the model for feature extraction.

## 2.4 MULTI-SUBJECT SHARED NEURAL RESPONSE REPRESENTATION

To capture the distinct processing patterns of visual stimuli across different subjects and encode them into the low-level and high-level token representations $(\mathbf{z}_{llv}, \mathbf{z}_{hlv})$, CLIP-MUSED leverages the topological relationships among visual stimuli in the feature spaces $\mathcal{F}_{llv}$ and $\mathcal{F}_{hlv}$ of CLIP to guide the representation learning process.

The guidance is realized by representational similarity analysis (RSA). First, $B$ neural signals are randomly sampled to obtain the low-level and high-level representations $(\mathbf{z}_{llv}, \mathbf{z}_{hlv})$. Next, we compute the representational similarity matrices (RSMs) $\mathbf{M}_{llv}^{\mathbf{X}}, \mathbf{M}_{hlv}^{\mathbf{X}} \in R^{B \times B}$ for the low-level and high-level representations $\mathbf{z}_{llv}$ and $\mathbf{z}_{hlv}$, respectively. Here, $\mathbf{M}_{llv}^{\mathbf{X}}[i, j]$ represents the cosine similarity between $\mathbf{z}_{llv}[i]$ and $\mathbf{z}_{llv}[j]$. During training, we shuffle the samples of all subjects and randomly sample from them. In a mini-batch, $\mathbf{z}_{llv}[i]$ and $\mathbf{z}_{llv}[j]$ may come from different subjects. The topological relationships of visual stimuli in the multimodal feature space and the shared space are aligned by minimizing the squared F-norm of the difference matrix normalized by the matrix size, i.e.,

$$\mathcal{L}_{llv} = \left\|\mathbf{M}_{llv}^{\mathbf{I}} - \mathbf{M}_{llv}^{\mathbf{X}}\right\|_F^2 / B^2, \tag{10}$$

$$\mathcal{L}_{hlv} = \left\|\mathbf{M}_{hlv}^{\mathbf{I}} - \mathbf{M}_{hlv}^{\mathbf{X}}\right\|_F^2 / B^2. \tag{11}$$

In addition to RSA, we can also establish the association between visual stimuli and neural signal representations using a mapping network. However, RSA-based method is superior to the mapping-based method. We offer a detailed explanation of why we chose RSA in Section B of the supplementary materials.

## 2.5 SEMANTIC CLASSIFIER

Both low-level visual features and high-level semantic features are crucial for semantic classification. To leverage both types of features, the low-level and high-level token representations are concatenated and fed into an MLP network for classification. The model outputs predicted probabilities $\hat{\mathbf{y}}$. The semantic classification process can be formalized as follows:

$$\mathbf{z} = \text{CONCAT}(\mathbf{z}_{llv}, \mathbf{z}_{hlv}), \tag{12}$$

$$\hat{\mathbf{y}} = \text{MLP}(\mathbf{z}). \tag{13}$$

The cross-entropy loss function is used as the classification loss.

$$\mathcal{L}_c = -\frac{1}{C} \sum_{j=1}^{C} \left[ \mathbf{y}_j \log(\hat{\mathbf{y}}_j) + (1 - \mathbf{y}_j) \log(1 - \hat{\mathbf{y}}_j) \right]. \tag{14}$$

## 2.6 OPTIMIZATION OBJECTIVE

To encourage low-level and high-level token representations for each stimulus to differ as much as possible, the proposed method applies an orthogonal constraint

$$\min \mathcal{L}_\perp = \left\| \mathbf{z}_{llv} \mathbf{z}_{hlv}^T \right\|_F^2 / B^2. \tag{15}$$

The optimization objective of the method is

$$\min \mathcal{L} = \mathcal{L}_c + \lambda_\perp \mathcal{L}_\perp + \lambda_{llv} \mathcal{L}_{llv} + \lambda_{hlv} \mathcal{L}_{hlv}, \tag{16}$$

where $\lambda_\perp, \lambda_{llv}, \lambda_{hlv}$ are trade-off parameters.

## 3 EXPERIMENTS

### 3.1 DATASETS

**HCP** (Glasser et al., 2013; Van Essen et al., 2012): This dataset is a part of the Human Connectome Project (HCP), containing BOLD signals from 158 subjects. To reduce computational demands, we randomly select nine subjects for subsequent experiments. The visual stimuli consist of four dynamic movie clips, each annotated with an 859-dimensional WordNet label (Miller, 1995). The top 53 categories with a frequency higher than 0.1 are selected for subsequent experiments.

**NSD (Natural Scenes Dataset)** (Allen et al., 2022): The dataset contains BOLD signals from eight subjects. The visual stimuli consist of natural images from the MSCOCO dataset (Lin et al., 2014), each with multiple labels from 80 categories. Each subject viewed a total of 10,000 stimuli. Unlike subjects in HCP, subjects in NSD viewed different stimuli. Of the 10,000 stimuli, 9,000 have no overlap between subjects, while the remaining 1,000 stimuli were presented to all subjects. However, some subjects did not complete all sessions, and some trials are not publicly available. For this study, the number of stimuli per subject is approximately 9,000.

A detailed description on how we preprocess the data to adapt the input form of Transformer, split the dataset, and obtain the multimodal features of each stimulus is provided in Section A of the supplementary materials.

### 3.2 BASELINE METHODS

To validate the effectiveness of the proposed method, we compare it with single-subject decoding methods and existing multi-subject decoding methods.

**Single-subject decoding methods:** These are methods using single-subject neural signals as input and classification loss as the constraint. **SS-MLP**, **SS-CNN**, and **SS-ViT** employ MLP, 3D-CNN, and ViT as the backbone networks, respectively. Specifically, SS-MLP and SS-CNN are used for the NSD and HCP datasets, respectively.

**Multi-subject data aggregation methods:** These methods train a single decoding model for all subjects by direct data aggregation. We refer to these methods as MS-SMODEL-MLP, MS-SMODEL-CNN, and MS-SMODEL-ViT, which use MLP, 3D-CNN, and ViT as the backbone networks, respectively.

**MS-EMB (Chehab et al., 2022):** MS-EMB trains a single model for all subjects by direct data aggregation. In contrast to MS-SMODEL, MS-EMB learns a token for each subject to encode their identity. This approach is based on a method proposed by Chehab et al. (2022) and uses ViT as the backbone network.

**Shared response model (SRM) (Chen et al., 2015):** SRM is a probabilistic generative hyperalignment method. However, SRM cannot handle the case where subjects view different stimuli, so it is only used on the HCP dataset. We use the open-source code available in BrainIAK. We search the optimal dimensionality of the shared latent space in intervals of 100 dimensions within the range of [100, 600]. The optimal dimensionality is 500.

### 3.3 PARAMETER SETTINGS

For the HCP dataset, we first extract volume features using a six-layer 3D-CNN (Dai et al., 2022) as shown in Fig. 3(a). This results in features of size $7 \times 8 \times 7 \times 512$, which are then reshaped into features of size $392 \times 512$. Each 512-dimensional vector serves as a patch for the Transformer. Afterward, CNN and Transformer layers are alternated. CLIP-MUSED employs a two-layer Transformer. Note that all CNN and Transformer layers are trained together. The learning rate is set to 0.001, and the batch size is 64, and the optimizer is Adam. We find the optimal values for the hyperparameters $\lambda_\perp$, $\lambda_{hlv}$, and $\lambda_{llv}$ by grid-search within the range of [0.001, 0.01, 0.1] and the best values are $\lambda_\perp = 0.001$, $\lambda_{hlv} = 0.001$, $\lambda_{llv} = 0.1$. The models converge after approximately three hours of training on one NVIDIA A100 GPU.

For the NSD dataset, the network structure of Transformer is similar to that of ViT (Dosovitskiy et al., 2021), with a patch embedding dimension of 512, 24 layers, and 8 heads for multi-head self-attention. The learning rate is set to 0.0001, the batch size is 64, and the optimizer is Adam. We find the optimal values for the hyperparameters $\lambda_\perp$, $\lambda_{hlv}$, and $\lambda_{llv}$ by grid-search within the range of [0.0001, 0.001, 0.01] and the best values are $\lambda_\perp = 0.001$, $\lambda_{hlv} = 0.001$, $\lambda_{llv} = 0.0001$. The models converge after approximately two hours of training on one NVIDIA A100 GPU.

### 3.4 EVALUATION METRICS

We employ three commonly used evaluation metrics in the field of multi-label classification: mean Average Precision (mAP), the area under the receiver operating characteristic curve (AUC) and Hamming distance.

### 3.5 COMPARATIVE EXPERIMENTAL RESULTS

Table 1 presents the results on the HCP dataset. Firstly, our method outperforms two single-subject decoding methods, SS-CNN and SS-ViT. SS-ViT and our method have the same backbone models, yet our method achieves significantly better metrics than SS-ViT, highlighting the superiority of the efficient data aggregation strategy of our method. The comparisons of SS-ViT and our method on different subjects are shown in Fig. C5 in the supplementary materials. Secondly, our method is also highly competitive when compared to the other multi-subject decoding methods (MS-SMODEL-CNN, MS-SMODEL-ViT, MS-EMB, and SRM). Despite having the same training data and backbone models, our proposed method outperforms MS-SMODEL-ViT on all metrics. This indicates that the subject-specific tokens employed in our method can handle individual differences well and are superior to the simple aggregation of multi-subject data used in MS-SMODEL-ViT.

The method effectiveness is further validated on the NSD dataset, where the stimuli in the training set for each subject are completely exclusive. Table 2 presents the results, which show that our

Table 1: Performance of different methods on the HCP dataset. All the improvement of our method compared to other methods is significant ($t$-test, $p < 0.05$) except for those underlined, where the p-values have been corrected with the Holm-Bonferroni method for multiple comparisons.

| Methods | mAP ↑ | AUC ↑ | Hamming ↓ |
|---|---|---|---|
| SS-CNN | $.347 \pm .001$ | $.531 \pm .006$ | $.351 \pm .017$ |
| SS-ViT | $.360 \pm .002$ | $.556 \pm .004$ | $.333 \pm .021$ |
| MS-SMODEL-CNN | $.342 \pm .001$ | $.525 \pm .004$ | $.528 \pm .091$ |
| MS-SMODEL-ViT | $.367 \pm .003$ | $\underline{.576 \pm .001}$ | $.310 \pm .014$ |
| MS-EMB | $.368 \pm .004$ | $.572 \pm .004$ | $.305 \pm .003$ |
| SRM | $.348 \pm .002$ | $.534 \pm .003$ | $\mathbf{.204 \pm .000}$ |
| **OURS** | $\mathbf{.373 \pm .002}$ | $\mathbf{.581 \pm .008}$ | $.283 \pm .004$ |

Table 2: Performance of different methods on the NSD dataset. All the improvement of our method compared to other methods is significant ($t$-test, $p < 0.05$), where the p-values have been corrected with the Holm-Bonferroni method for multiple comparisons.

| Methods | mAP ↑ | AUC ↑ | Hamming ↓ |
|---|---|---|---|
| SS-MLP | $\mathbf{.258 \pm .005}$ | $.854 \pm .004$ | $.033 \pm .001$ |
| SS-ViT | $.238 \pm .005$ | $.815 \pm .008$ | $.032 \pm .000$ |
| MS-SMODEL-MLP | $.150 \pm .005$ | $.767 \pm .006$ | $.039 \pm .001$ |
| MS-SMODEL-ViT | $.156 \pm .006$ | $.755 \pm .014$ | $.038 \pm .001$ |
| MS-EMB | $.220 \pm .014$ | $.825 \pm .030$ | $.035 \pm .001$ |
| **OURS** | $\mathbf{.258 \pm .017}$ | $\mathbf{.877 \pm .021}$ | $\mathbf{.030 \pm .002}$ |

Table 3: Results of the ablation study on the NSD dataset. Our method outperforms the methods with other loss configurations significantly ($t$-test, $p < 0.05$), where the p-values have been corrected with the Holm-Bonferroni method for multiple comparisons.

| $\lambda_{\perp}$ | $\lambda_{hlv}$ | $\lambda_{llv}$ | mAP ↑ | AUC ↑ | Hamming ↓ |
|---|---|---|---|---|---|
| | ✓ | ✓ | $.201 \pm .006$ | $.794 \pm .010$ | $.036 \pm .001$ |
| ✓ | | | $.209 \pm .002$ | $.819 \pm .009$ | $.036 \pm .002$ |
| ✓ | | ✓ | $.215 \pm .012$ | $.820 \pm .018$ | $.036 \pm .001$ |
| ✓ | ✓ | | $.209 \pm .016$ | $.824 \pm .017$ | $.037 \pm .002$ |
| ✓ | ✓ | ✓ | $\mathbf{.258 \pm .017}$ | $\mathbf{.877 \pm .021}$ | $\mathbf{.030 \pm .002}$ |

method outperforms the single-subject methods, SS-MLP and SS-ViT. The comparisons of SS-ViT and our method on different subjects are shown in Fig. C6 in the supplementary materials. With the same amount of data and the same backbone models, the aggregation methods are far inferior to our method. This is mainly because the MS-SMODEL methods sharing all the model parameters across all subjects are hard to handle both inter-subject variability and differences in stimuli distribution. Although MS-EMB performs better than the aggregation methods, it is still inferior to the proposed method.

## 3.6 ABLATION STUDY

We conduct an ablation study on the NSD dataset, and the results are shown in Table 3. The model performance is unsatisfactory when only model guidance is applied without orthogonal constraints or when only orthogonal constraints are applied without model guidance. Using either low-level or high-level features for guidance with orthogonal constraints results in a slight improvement in model performance, but there is still a gap compared to the model's performance with all three constraints. These results confirm the necessity of multimodal model guidance and orthogonal constraints on primary and high-level token representations. We also investigate the guidance effect of CLIP on SS-ViT and the guidance effect of different DNNs. The results are shown in Table D5 and Table E6 in the supplementary materials.

## 3.7 VISUALIZATION

We visualize the attention maps of low-level and high-level tokens on the last layer of CLIP-MUSED model. The visualization results on the left hemispheres of HCP dataset are presented in Fig. 4, where we randomly select four subjects for presentation. Fig. 4(a) shows the attention maps for the low-level tokens, which are concentrated in the occipital lobe. Fig. 4(b) shows the attention maps of the high-level tokens, which are more dispersed across the cortex, with strong attention allocated to the frontal, parietal, and temporal lobes. These results are in line with our expectations, as previous studies have demonstrated that the processing of low-level visual features mainly occurs

in the visual cortex, while the processing of high-level semantic features involves the temporal, parietal, and frontal lobes (De Benedictis et al., 2014; Mitchell et al., 2008). Fig. 4(c) shows the attention maps of the embedding tokens of MS-EMB. In contrast to the token attention maps of CLIP-MUSED, the attention maps of MS-EMB are smoothly distributed across the entire cortical surface, making it difficult to understand which information is encoded in the tokens.

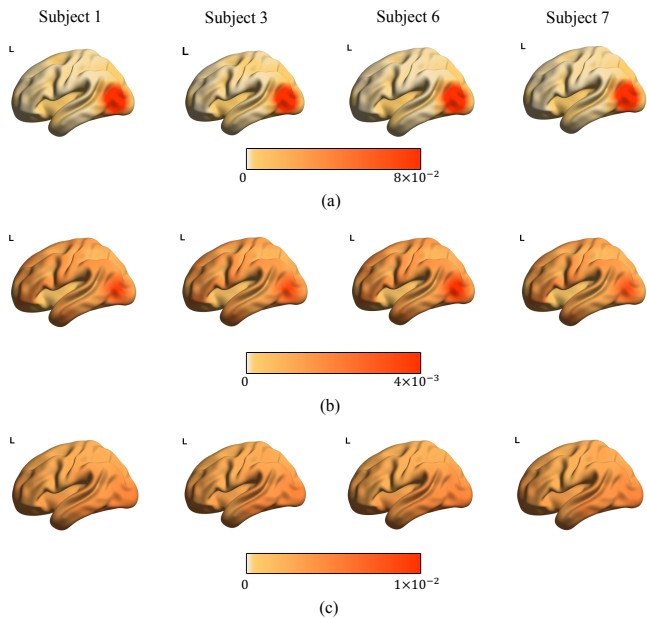

Figure 4: On the HCP dataset, attention maps of (a) low-level tokens and (b) high-level tokens of our method, and (c) attention maps of subject embeddings of the MS-EMB method, were visualized on the cortical surface for 4 randomly selected subjects.

The visualized results on the NSD dataset are displayed in Fig. F8 of the supplementary materials. The results provide explanations for the superior performance of the proposed method.

## 4 DISCUSSION AND CONCLUSION

To address the issue of individual variability, we propose a multi-subject semantic decoding method, CLIP-MUSED. We design a Transformer-based encoder that can extracts global features of neural responses. The method introduces subject-specific low-level and high-level tokens to encode individual variability. Based on the RSA, CLIP-MUSED guides the representation learning of tokens by the topological relationship of visual stimuli in the representation space of CLIP. On two fMRI datasets, the proposed method outperforms the single-subject methods and achieves state-of-the-art performance.

All parameters of CLIP-MUSED, except for the subject-specific tokens, are shared among all subjects. This allows the method to be extended to datasets with hundreds of subjects. Moreover, CLIP-MUSED is applicable to situations where multiple subjects have different stimuli. Even when the stimulus images of different subjects are completely mutually exclusive, our method outperforms single-subject decoding models. Therefore, the proposed method has the potential to train a foundation model with aggregating multi-source neural datasets, following the research trends in the computer vision and natural language processing communities (Radford et al., 2021; Brown et al., 2020).

In the future, we plan to extend our method to visual stimuli reconstruction (Chen et al., 2022; Lin et al., 2022; Takagi & Nishimoto, 2022), a more challenging task than semantic classification. Besides, our approach can be applied to new subjects if we devise strategies to learn subject-specific tokens for new subjects and adapt the model accordingly. Since the experimental workload are substantial, we plan to carry out this work in the future.

## 5 ACKNOWLEDGEMENTS

This work was supported in part by the National Key R&D Program of China (2023YFF1203500); in part by the National Natural Science Foundation of China under Grant 62206284, Grant 62020106015 and Grant 61976209; and in part by the CAAI-Huawei MindSpore Open Fund.

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

## A  DATA PREPROCESSING

**Preprocessing before input:** The original BOLD volumes in the HCP dataset have dimensions of $113 \times 136 \times 113$. The 3-D BOLD volumes are patchified by 3D-CNN as shown in Fig. 3(a) during experiments. For the NSD dataset, we extract ROIs in the visual cortex based on brain parcellation, including V1-V3, V3ab, hV4, ventral occipital (VO), intraparietal sulcus (IPS), lateral occipital (LO), middle temporal (MT), and parahippocampal cortex (PHC), covering both primary and high-level areas of the visual cortex. The number of voxels in each ROI varies across subjects, and the number of voxels for the same subject also varies across different ROIs. To enable model weight sharing across subjects, we align the dimensions of the BOLD signals in the same ROI across subjects. Additionally, to adapt the data to the Transformer model, we align the dimensions of the BOLD signals in the same subject across different ROIs. To achieve this, we employ principal component analysis to reduce the dimensionality of the BOLD signals to the minimum dimensionality value of 268 across all BOLD signals from all ROIs of all subjects. During the testing process, we can apply zero-padding to neural signals with dimensions lower than this value. In the methods using MLP as the backbone network, the patches of all ROIs are concatenated into a vector to input into the model. In the methods using ViT as the backbone network, one patch is one token.

**Data split:** The HCP dataset is split in accordance with the method described in Khosla et al. (2020). The first three movies are used for training and validation, while the fourth movie is used for test. For the single-subject decoding task, the training, validation, and test sets consist of 2000, 265, and 699 samples, respectively. For the multi-subject decoding task, the training sets of nine subjects are combined and randomly shuffled for model training, while the validation sets of the same nine subjects are combined for model validation. To account for the randomness, we report the average results of three random runs. A hemodynamic delay of 4 seconds estimated in Khosla et al. (2020) is used in this paper. For the NSD dataset, the stimuli viewed by all subjects and their corresponding neural responses are used for test in the single-subject decoding task. A validation set consisting of 1,000 randomly sampled examples is used for hyper-parameter tuning and convergence monitoring during training. The remaining data is used for training. For the multi-subject decoding task, the training sets of eight subjects are combined and randomly shuffled, and a validation set of 1,000 examples is randomly sampled. The remaining data is used for training. Due to the randomness of data split, we report the average results of five random splits.

**Multimodal feature extraction:** We concatenate the WordNet annotations of each movie frame to form the textual information for stimuli in the HCP dataset. In the NSD dataset, each stimulus image is associated with five captions. Following the approach in Lin et al. (2022), we compute the similarity between each image and its five captions in the multimodal feature space of CLIP. We use half of the maximum similarity score as the threshold and randomly select one caption from the selected candidates to serve as the textual information for the stimulus. We extract the text features by inputting the textual information into the CLIP text encoder. For both datasets, we truncate the image and text features with a threshold of 1.5 and normalize their L2 norms to 1. We then compute the average of the image and text features to obtain the multimodal feature $\mathbf{f}_{hlv}$, which guides the learning of $\mathbf{z}_{hlv}$.

## B  THE CHOICE OF AN RSA-BASED LOSS

Using RSA to guide the representation learning of fMRI is superior to directly mapping fMRI representations to the CLIP embeddings. There are three main reasons:

1. Mapping-based methods tend to focus on learning local information when embedding fMRI representations into the CLIP space (e.g., minimizing the L2 norm between true and predicted values). However, they may overlook the global topological structure of the CLIP space, which can affect the interpretability and generalization capabilities of the embedding space. In contrast, using RSA loss ensures the global topological structure similarity between the embedding space and the CLIP space.

2. There is a gap between the representation spaces of CLIP and fMRI. It is more challenging to directly learn the mapping from fMRI representations to CLIP representations than to learn the topological relationship. Forcing alignment may lead to overfitting and poor generalization.

Table B4: Results of the RSA-based method (CLIP-MUSED) and mapping-based methods on the NSD dataset.

| Method | mAP ↑ | AUC ↑ | Hamming ↓ |
|---|---|---|---|
| Mapping-Based | $.247 \pm .026$ | $.844 \pm .033$ | $.035 \pm .002$ |
| RSA-Based | $\mathbf{.258 \pm .017}$ | $\mathbf{.877 \pm .021}$ | $\mathbf{.030 \pm .002}$ |

3. Mapping-based methods introduce additional trainable parameters in the mapping network, and the architecture of the mapping network also needs to be delicately optimized.

We also conduct a comparative experiment between RSA-based and mapping-based methods on the NSD dataset. The results in Table B4 demonstrate the superiority of the RSA-based loss. The coefficient ($\lambda = 0.0001$) of the mapping-based loss item has been optimized.

## C  PEFORMANCE COMPARISON ON EACH SUBJECT

We compare the performance of SS-ViT, which is a single-subject decoding method with CLIP-MUSED, as they both utilize ViT as the backbone network. Fig. C5 compares the mAP of our method and SS-ViT on each subject. As shown in the figure, our method outperforms SS-ViT on all subjects. Fig. C6 show the comparison results on the NSD dataset. It is evident that our method consistently outperforms SS-ViT on the majority of subjects. These findings suggest that the neural representations shared among subjects learned by our method are superior to those learned by the single-subject method.

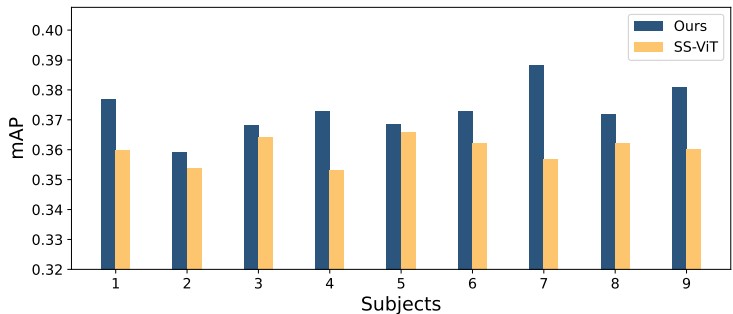

Figure C5: Performance comparison between SS-ViT and our method on each subject of the HCP dataset.

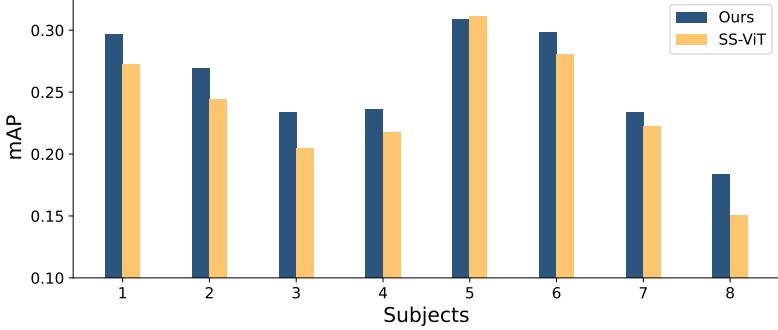

Figure C6: Performance comparison between SS-ViT and our method on each subject of the NSD dataset.

## D  GUIDANCE EFFECT ON THE SINGLE-SUBJECT METHOD

In the experiments, we only employ CLIP guidance in our proposed multi-subject model. To investigate the impact of CLIP guidance on the single-subject model, we modified the SS-ViT to

Table D5: Results of two single-subject methods CLIP-SS-ViT and SS-ViT, with and without CLIP guidance, and CLIP-MUSED on the NSD dataset. All the improvement of CLIP-MUSED compared to other cases is significant ($t$-test, $p < 0.05$), where the p-values have been corrected with the Holm-Bonferroni method for multiple comparisons.

| Method | mAP ↑ | AUC ↑ | Hamming ↓ |
|---|---|---|---|
| SS-ViT | $.238 \pm .005$ | $.815 \pm .008$ | $.032 \pm .000$ |
| CLIP-SS-ViT | $.234 \pm .002$ | $.822 \pm .006$ | $.032 \pm .001$ |
| CLIP-MUSED | $\mathbf{.258 \pm .017}$ | $\mathbf{.877 \pm .021}$ | $\mathbf{.030 \pm .002}$ |

enable the learning of neural representations under the guidance of CLIP, i.e., CLIP-SS-ViT. The model structure and composition of the loss function for CLIP-SS-ViT are the same as our method, with the only difference being that CLIP-SS-ViT is trained on single-subject data. We carefully tuned the trade-off parameters for CLIP-SS-ViT, and the optimal parameters are $\lambda_\perp = 0.0001$, $\lambda_{hlv} = 0.0001$, and $\lambda_{llv} = 0.0001$. The results are shown in Table D5.

It can be observed that, both guided by CLIP, our method outperforms the CLIP-SS-ViT. This superiority can be attributed to the multi-subject aggregation strategy employed in our method. On the single-subject model, the relatively weak effect of using CLIP as guidance may be due to the fact that the single-subject model does not need to handle individual differences. In other words, the original classification loss can implicitly learn the relationship between different fMRI representations of a single subject, and the guidance from CLIP is redundant for the model.

# E  GUIDANCE EFFECT OF DIFFERENT DNNS

Table E6 presents the model performance when the neural representation learning is guided by the topological relationship of visual stimuli across different DNN representation spaces. CLIP-Img/CLIP-Text refer to the utilization of high-level image/text features extracted by the image/text encoder of CLIP, instead of the multimodal features, during the learning of high-level tokens. In summary, the results depicted in Table E6 suggest that features extracted from CLIP exhibit a superior guidance effect when compared to the baseline models (ViT and AlexNet). Notably, textual information provides a richer semantic understanding of the visual stimuli and integrating it with the image features can serve to augment the performance of the model in guiding neural representation learning.

Table E6: Comparison of the guidance effect of different DNNs. All the improvement of CLIP compared to other cases is significant ($t$-test, $p < 0.05$) except for those underlined, where the p-values have been corrected with the Holm-Bonferroni method for multiple comparisons.

| Methods | HCP | | | NSD | | |
|---|---|---|---|---|---|---|
| | mAP ↑ | AUC ↑ | Hamming ↓ | mAP ↑ | AUC ↑ | Hamming ↓ |
| ViT | $.362 \pm .003$ | $.562 \pm .004$ | $.383 \pm .040$ | $.229 \pm .009$ | $.849 \pm .014$ | $.035 \pm .001$ |
| AlexNet | $.362 \pm .002$ | $.558 \pm .005$ | $.333 \pm .013$ | $.225 \pm .012$ | $.855 \pm .013$ | $.034 \pm .002$ |
| CLIP-Img | $.370 \pm .001$ | $\underline{.577 \pm .001}$ | $.291 \pm .002$ | $.238 \pm .006$ | $.863 \pm .008$ | $.033 \pm .001$ |
| CLIP-Text | $.370 \pm .002$ | $\underline{.579 \pm .006}$ | $.286 \pm .003$ | $.223 \pm .006$ | $.841 \pm .008$ | $.036 \pm .001$ |
| **CLIP (OURS)** | $\mathbf{.373 \pm .002}$ | $\mathbf{.581 \pm .008}$ | $\mathbf{.283 \pm .004}$ | $\mathbf{.258 \pm .017}$ | $\mathbf{.877 \pm .021}$ | $\mathbf{.030 \pm .002}$ |

# F  VISUALIZATION

We present the visualization of between-subject representational similarity matrices (RSMs) of low-level and high-level tokens on two datasets, as depicted in Fig. F7. Notably, the similarity of tokens between subjects is observed to be higher on the HCP dataset compared to the NSD dataset. This can be attributed to the fact that in the HCP dataset, all subjects viewed the same stimuli, and the distribution of stimuli across subjects is uniform, whereas in the training set of the NSD dataset, the stimuli viewed by different subjects are mutually exclusive. Evidently, different subjects process

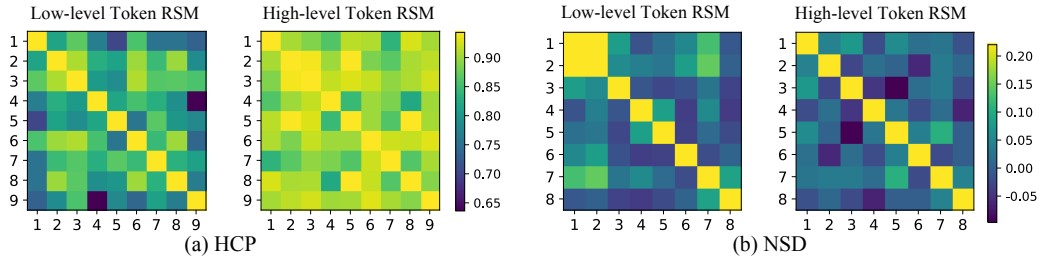

Figure F7: RSM between low-level and high-level tokens across subjects of our method on (a) the HCP dataset and (b) the NSD dataset.

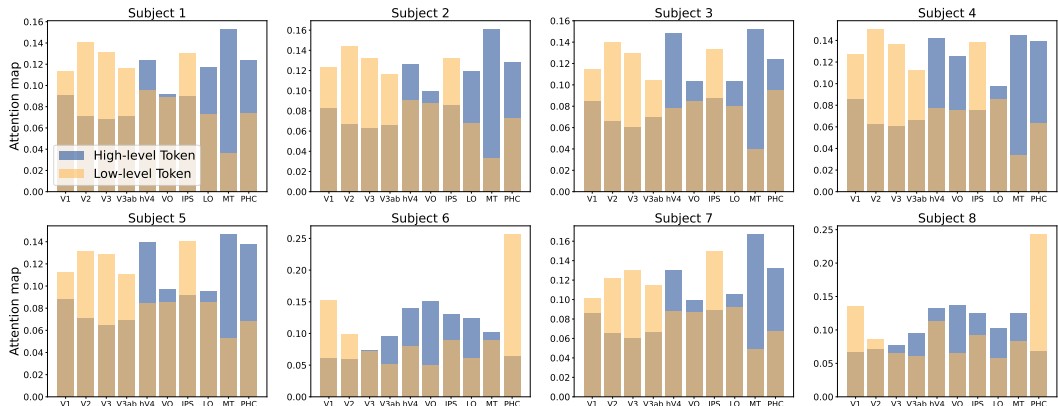

Figure F8: Attention maps between the low-level and high-level tokens of our method and different brain region tokens at the last Transformer self-attention layer for 8 subjects on the NSD dataset.

stimulus information in slightly distinct patterns even when viewing the same stimuli, and these differences are further amplified when presented with different stimuli. The tokens learned for each subject in CLIP-MUSED can encode these inter-subject variabilities, resulting in lower similarity of tokens between subjects on the NSD dataset with different stimuli for different subjects.

Fig. F8 shows the attention maps of the low-level and high-level tokens on different ROIs of the NSD dataset. The low-level tokens exhibit a higher attention allocation towards the primary and intermediate visual cortex regions on the left side of the bar chart, including V1-V4, while the high-level tokens exhibit a more pronounced attention towards the higher visual cortex regions such as LO, MT, and PHC. Prior research has established that MT plays a crucial role in depth perception (Born & Bradley, 2005), LO is involved in object recognition tasks (Grill-Spector et al., 2001), and PHC contributes to visual perception related to memory and spatial scenes (Aminoff et al., 2013). In contrast, Fig. F9 demonstrates the attention maps of the subject embeddings in the MS-EMB method, revealing a more focused attention on the V3, V4, and VO brain regions, but less on the higher visual cortex regions such as LO, MT, and PHC. In contrast, our method leverages both low-level and high-level tokens to allocate attention towards both low-level and intermediate visual cortex regions, as well as higher visual cortex regions. The difference in token attention maps between our method and the MS-EMB method partially elucidates the superiority of our method in classification performance.

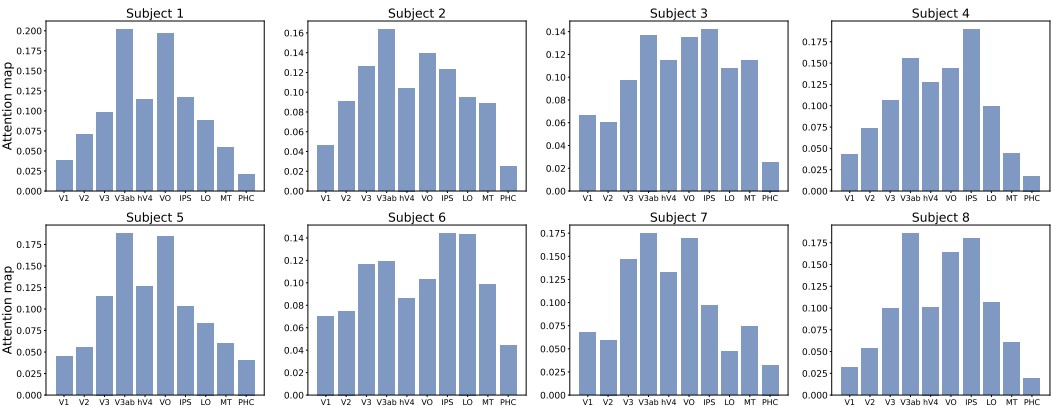

Figure F9: Attention maps between the subject embedding token of the MS-EMB method and different brain region tokens at the last Transformer self-attention layer for 8 subjects on the NSD dataset.

