# OpenReview forum: "CLIP-MUSED: CLIP-Guided Multi-Subject Visual Neural Information Semantic Decoding"
_ICLR.cc/2024/Conference — ICLR 2024 poster_

### Official Review · Reviewer_UuaC · 2023-10-31

**Soundness:** 3 good
**Presentation:** 3 good
**Contribution:** 3 good
**Rating:** 5
**Confidence:** 4

**Summary:**

This paper proposes a CLIP-guided Multi-sUbject visual neural information SEmantic Decoding (CLIP-MUSED) method which generalizes single-subject decoding models to multiple subjects in visual neural decoding tasks. Different from other multi-subject decoding methods, CLIP-MUSED uses a Transformer-based fMRI feature extractor to effectively extract global features of neural responses. CLIP-MUSED uses low-level and high-level tokens to encode individual differences and uses the topological relationship of visual stimuli in CLIP representation space based on RSA to guide the representation learning of tokens.

**Strengths:**

1. The paper is easy to read, and generally well written.
2. The use of RSA is novel and contributes to the topological relationships of visual stimuli in the CLIP representation space as prior knowledge to guide neural representation learning in the shared space.
3. The experiments have clearly verified the benefits of the proposed methods.

**Weaknesses:**

1. The use of letters in the article needs to be consistent. For example, the representation of X.
2. In order to reduce the computational cost, using 3D-CNN to reduce the dimension will have an impact on the results? The results of this comparison are not shown.
3. The high-level feature in RSMs uses the average of the image and text features from the last layer of CLIP. What are the advantages of using the CLIP image encoder or text encoder alone compared to directly using it alone? Can you provide corresponding results for an explanation?

**Questions:**

1. How to set labels for semantic classification of N subjects after concatenating low-level and high-level token representations of different image stimuli?
2. In Section 2.1 "X^{(n)}" is recommended to be consistent with the previous notation. The rest of the article should also be consistent.
3. The meaning of B in "B visual stimuli" in Section 2.2 should be explained.
4. Is the underline in the AUC results of MS-SMODEL-ViT in Table 1 redundant? It should remain in the same format as other tables.

---

> ### Author Response · Authors · 2023-11-14
> **Response_1-Part I**
>
> We are grateful for your constructive comments! Here are our responses:
>
>
> **Response to the mentioned weaknesses**:
>
> **W1: Letter consistency**:
>
> The expressions $\mathcal{X}^{(n)}$ and $\mathbf{X}^{(n)}$ actually represent two different meanings. $\mathcal{X}^{(n)}$ represents the neural response space of the n-th subject, while $\mathbf{X}^{(n)}$  represents the samples in the dataset, which is a subset of the former. We also checked the other symbols.
>
> **W2: The impact of 3D-CNN**：
>
> In previous studies, 3D-CNN has frequently been employed for extracting features from volumetric BOLD signals, including works in neural encoding/decoding [1] and medical image processing [2][3]. Even in recent studies that introduced Transformers [4][5], 3D-CNN is still utilized at the model's foundation for dimensionality reduction and feature extraction. These investigations emphasize not only the computational efficiency of CNN but also its capability to smooth out noise in BOLD signals.
>
> During the initial experiments of this study, we attempted to directly input volumetric BOLD signals, split into patches, into a Transformer. Specifically, we divided the signals into 6x7x6 patches, each containing $19^3$ voxels. Despite employing a 24-layer ViT with an embedding size of 1024, the loss function on the training set does not exhibit a convergent trend during training. This suggests that directly inputting responses in the voxel-space into a Transformer is less feasible, while preliminary feature extraction through 3D-CNN contributes to stable convergence and enhanced classification performance.
>
> We refrained from extensive optimization on the network architecture of the 3D-CNN used in our model and instead drew inspiration from the study in [5] buiding a model composed of 3D-CNN and Transformer. In the experimental phase, we maintain consistency in the CNN architectures used by all baseline methods based on 3D-CNNs (except for method-specific modules), ensuring a fair comparison to validate the effectiveness of our approach.
>
>
> **W3: Results of single-modal feature guidance**：
>
> Although CLIP maximizes the similarity between image and text modal embeddings through contrastive learning, there still exist differences in the last-layer embeddings of images and text. By considering the embeddings from both modalities and averaging them, the model can obtain more information.
>
> We also conduct related experiments. As you can see, Table E6 shows the results only using the last-layer features extracted by CLIP's image encoder to guide the learning of higher-level representations, which is not better than CLIP-MUSED guided by the multi-modal features.
> The experimental results using only the last-layer features from CLIP's text encoder for guidance have also been added to Table E6 (See CLIP-Text). These results further highlight the advantages of using the RSM of the multimodal features as the guidance.

---

> > ### Author Response · Authors · 2023-11-14
> > **Response_1-Part II**
> >
> > **Q1:Label settings for semantic classification**
> >
> > **response**: The labels for the stimulus images are represented as a C-dimensional vector, where C is the number of label categories. Within this vector, 1/0 indicates the presence/absence of an object category in the stimulus image. Given the fMRI response of a sample, low-level and high-level token representations are firstly obtained and concatenated. Subsequently, the MLP in Eq. 13 maps the concatenated representation to an C-dimensional space. The model outputs an C-dimensional logits vector, and each dimension undergoes a sigmoid activation function transformation, yielding predicted probabilities ($p_i$) for each label. If $p_i$ > 0.5, the model predicts the presence of label $i$; if $p_i$ < 0.5, the model predicts the absence of label $i$.
> >
> > **Q2: Letter consistency**
> >
> > **response**: Please see our response to W1.
> >
> > **Q3. The meaning of B in "B visual stimuli" in Section 2.2.**
> >
> > **response**: The symbol B in "B visual stimuli" represents the batch size in the mini-batch training paradigm. We have provided the corresponding details in Section 2 as indicated.
> >
> > **Q4. The underline in the AUC results of MS-SMODEL-ViT in Table 1.**
> >
> > **response**: We clarify that the underlines in the data of Table 1 and Table E6 are not redundant. As indicated in the captions of these two tables, the underlines signify that the improvement of our method compared to the reference methods is not significant in the respective metric.
> >
> > **References**
> >
> > [1] Vu, et al. (2020). fMRI volume classification using a 3D convolutional neural network robust to shifted and scaled neuronal activations. NeuroImage.
> >
> > [2] Parmar, et al. (2020). Deep learning of volumetric 3D CNN for fMRI in Alzheimer's disease classification. Medical Imaging 2020: Biomedical Applications in Molecular, Structural, and Functional Imaging.
> >
> > [3] Oh, et al. (2019). Classification of schizophrenia and normal controls using 3D convolutional neural network and outcome visualization. Schizophrenia research.
> >
> > [4] Malkiel, et al. (2022). Self-supervised transformers for fMRI representation. International Conference on Medical Imaging with Deep Learning.
> >
> > [5] Dai, te al. (2022) BrainFormer: A hybrid CNN-Transformer model for brain fMRI data classification. arXiv:2208.03028.

---

> ### Author Response · Authors · 2023-11-21
>
> Dear Reviewer,
>
> We appreciate your valuable revisions and suggestions. In our response, we have taken the time to address your concerns and provide relevant clarifications. As the open discussion phase is drawing to a close, we are eagerly awaiting your feedback.
>
> Thank you for your time and effort in revising our paper.
>
> Best regards,
>
> Submission6614 Authors

---

### Official Review · Reviewer_qr5q · 2023-10-31

**Soundness:** 3 good
**Presentation:** 2 fair
**Contribution:** 2 fair
**Rating:** 6
**Confidence:** 3

**Summary:**

The authors propose a transformer based cross-subject fMRI alignment method that uses learned special tokens (an approach also used in other ViT/language works which perform classification).

Specifically, the paper discusses the following issues in traditional hyperalignment approaches:
1. Mapping functions are restrictive (I assume this is in reference to orthogonal procrustes, linear mappings, or kernelized mappings)
2. Requirements for a per-subject mapping function, which can be expensive when deep networks are used
3. Need for identical stimuli or stimuli from the same semantic category

They propose the following approach:
1. Using a transformer to model long-range dependencies
2. Using special tokens to model subject-wise information
3. CLIP distances to guide the model, with the average of the text & visual branches used for the high level feature.

**Strengths:**

CLIP-MUSED is an interesting approach that shares model parameters and only varies a per-subject token. The use of CLIP for feature extraction and feature alignment is solid. The method allows for the use of multi-subject data without a linear increase in the number of model parameters, which is beneficial for scalability

The paper provides a detailed methods section that covers the technical aspects of the CLIP-MUSED, including the use of Transformers and CLIP and the design of an RSA based loss. The high level approach is clear.

The paper is well-organized, making it relatively easy to follow the arguments and understand the methodology.

**Weaknesses:**

**In my view there are two weaknesses of the paper:**
1. The relatively weak decoding the authors perform, which is implied to be just category decoding. It is very strange to me that they use so much compute relative to traditional methods, and end up just doing category decoding. I believe the results would be strengthened by performing visual decoding in the form of full image reconstruction, at least for NSD (single image passive viewing task). If this is not possible, I think an alternative would be perform image retrieval (show top-5) on a test set, where the test set contains images not used for the training of any subject. This would likely require training the decoder on a single subject's stimuli only, then testing on other subjects.
2. The fact that the subject-wise token is not computed via amortized inference (via an encoder conditioned on a few BOLD responses). But instead, they are fixed per subject. In my view this weakens the approach somewhat as you cannot take this approach and apply it to a new subject in a few-shot fashion, after the paper talks about how their method does not require subjects to view the same stimulus.

**Math typos:**
1. In both the paper eq 10, the authors discuss using the F-norm (frobenius norm), but in practice they are using the squared frobenius norm in the code (line 42 of `losses.py`). This is fine, as the squared F-norm is everywhere differentiable, but I ask the authors clarify/fix this.
2. In the paper eq 15, the authors apply a orthogonal constraint such that matrix $z_\text{llv}$ and matrix $z_\text{hlv}$ are orthogonal to each other via the regularization loss. Same issue listed above applies. You should probably also clarify here that you want the low/high level representations for each stimuli to be orthogonal, otherwise it is not super clear.

**Minor typos:**
1. Page 3. Original `The contributes are summarized as follows`, should be `The contributions are summarized as follows`
2. Page 3. It seems that $X^{(n)} \in \mathbb{R}^{n_i \times d} $ should be $\mathcal{X}^{(n)} \in \mathbb{R}^{n_i \times d} $

Overall I think the proposed method is interesting, but the experiments currently do not fully demonstrate the strength of the method. I would gladly reconsider if the authors can clarify my questions, and provide additional experiments to support their method.

**Questions:**

Questions:
1. For Figure 1 in the paragraph below you mention `two bird images t1 and t2` as well as `t1 and t2 are birds, while t3 (duck) and t4 (building blocks) are not`.

    **a**. I encourage the authors to clarify this, or use other examples in the figure and paper. Ducks should be birds.

2. In the paper, it is discussed that `Linear transformation and MLP are unsuitable for high-dimensional voxel responses`. However no justification on the MLP aspect is given, and the citation (Yousefnezhad & Zhang) explicitly uses a kernelized MLP approach. Could you clarify?

3. In section 3.2, you never define what a backbone network is. Does it just take as input the fMRI BOLD responses and output a category? In that case, is the ViT a 3D model?

4. It is not clear what exactly you are decoding. You mention that HCP has a wordnet label, and NSD has the 80 categories (I assume the binary 0/1 mask on if an object from a category is present in the image).

5. Can you clarify if you use ROI based patching for NSD, and the CNN based patching approach for HCP? Is the CNN jointly trained with the transformer?

6. How do you extract visual features for HCP given that the stimulus set consists of video clips?

---

> ### Author Response · Authors · 2023-11-14
> **Response_1-Part I**
>
> We really appreciate your insightful review! Here are our responses:
>
> **Response to the mentioned weaknesses**:
>
> **W1: Category decoding task**:
>
> Thank you for your valuable suggestions. We are supplementing with retrieval experiments. The experimental results will be promptly released once the experiments are completed. Although reconstruction tasks also have significant research value, reconstruction involves deep generative models such as diffusion models, whereas our method is a discriminative model. Research on algorithms for reconstruction tasks is beyond the scope of this paper. We will consider conducting further research in the future.
>
> **W2: The learning of subject-wise token**:
>
> The amortized inference you mentioned is indeed a good idea, but our method also has advantages compared to the amortized inference.
> 1. Amortized inference typically relies on assumptions about the prior distribution. If the prior assumptions are inappropriate or inaccurate, it may affect the effectiveness of inference and the quality of results. Our method does not rely on prior distribution assumptions.
> 2. Our method eliminates the need for an encoding network, reducing the number of parameters that need to be learned and preventing overfitting.
>
> Applying to new subjects in a few-shot fashion still is a common challenge faced in the multi-subject neural decoding field. We have minimized the parameters that new subjects need to learn as much as possible.
>
> **W3: Math typos**
>
> In the latest manuscript PDF, we have revised the original statement to "the squared F-norm of the difference matrix normalized by the matrix size," and made the corresponding modification to Eq 10. Based on your suggestion, we have also modified the description of the orthogonal constraint, "To encourage the low-level and high-level token representations for each stimuli to represent as different features as possible, the proposed method applies an orthogonal constraint by ...".
>
> **W4: Minor typos**
> 1. In the latest manuscript PDF, we have corrected "Original The contributes are summarized as follows" to "The contributions are summarized as follows" in Page 3.
> 2. The expressions $\mathcal{X}^{(n)}$ and $\mathbf{X}^{(n)}$ actually represent two different meanings. $\mathcal{X}^{(n)}$ represents the neural response space of the n-th subject, while $\mathbf{X}^{(n)}$  represents the samples in the dataset, which is a subset of the former.

---

> > ### Author Response · Authors · 2023-11-14
> > **Response_1-Part II**
> >
> > **Q1: Duck image in Figure 1**
> >
> > **response**: Thank you for your suggestion. We have replaced the duck with cat.
> >
> >
> > **Q2: "Linear transformation and MLP are unsuitable for high-dimensional voxel responses."**
> >
> > **response**: First, we discussed that linear transformation is unsuitable for high-dimensional voxel responses because Yousefnezhad et al. pointed out in their paper (Yousefnezhad & Zhang) that "original hyperalignment does not work in a very high dimensional space". Second, since MLP is composed of stacked linear layers and activation functions, the multi-layer kernel function in (Yousefnezhad & Zhang) will have a large number of parameters. Based on this, the MLP-based method is not a parameter-efficient method and is not suitable for high-dimensional data. We have provided these explanations at the corresponding locations in the updated manuscript PDF.
> >
> > **Q3: Backbone network**
> >
> > **response**: The inputs for different backbones are preprocessed BOLD responses, and the outputs are class labels. The structure of the backbone network varies, and this variation also determines the differences in the format of the input data. In SS-MLP, the patchified responses of multiple ROIs are concatenated together, while in SS-ViT, the patchified responses are treated as different tokens. For HCP dataset, the model utilizes a 3D-CNN to convert 3D volumes to 2D patches, followed by a 2D ViT, as depicted in Figure 3(a)(b). The NSD dataset, on the other hand, directly employs 2D ViT, as shown in Figure 3(b). We have added these clarifications to Section 3.2 of the updated manuscript PDF and the section "Preprocessing before input" of Appendix A of the supplementary materials.
> >
> >
> > **Q4: Label format**
> >
> > **response**: In the HCP dataset, we utilize WordNet labels as semantic labels for the stimulus images. Each word in the WordNet is treated as a separate category or class. In NSD, the stimulus image labels are 80 categories. In both HCP and NSD, "1" indicates the presence of a certain category, and "0" indicates its absence. A single stimulus image may have multiple labels associated with it.
> >
> > **Q5: Data patching**
> >
> > **response**: We use ROI based patching for NSD, and the CNN based patching approach for HCP. Due to space limitation, we provide additional descriptions on how we process the NSD and HCP data to adapt the input form of Transformer in the section "Preprocessing before input" of Appendix A. In the HCP dataset, the CNN and transformer layers are trained jointly. We provide the clarification in Section 3.3 of the updated manuscript PDF.
> >
> > **Q6: Visual feature extraction**
> >
> > **response**: We referred to the setting of feature extraction for HCP video stimuli described in reference [1]. In HCP, the repetition time (TR) is 1 second. We extracted the features corresponding to each second of stimuli and combined them with the BOLD response occurring 4 seconds later, forming a sample. The choice of 4 seconds is based on the hemodynamic delay estimated in reference [1].
> >
> > **References**
> >
> > [1] Khosla et., al. A shared neural encoding model for the prediction of subject-speciﬁc fMRI response. MICCAI, 2020.

---

> > > ### Comment · Reviewer_qr5q · 2023-11-17
> > > **Thank you for the response**
> > >
> > > I'm writing to thank the authors for their clear and concise response.
> > >
> > > I have read the current revision, and I think broadly the clarifications are helpful.
> > >
> > > Here are a few minor typos introduced in the revision:
> > > * `Our method comprises a Transformer-based` -> `Our method is consists of a Transformer-based`
> > > * ` that facilitate the model to aggregate multi-subject data` -> `that facilitates the aggregation of multi-subject data`
> > > * `without linear increase` -> `without a linear increase`
> > > * `To encourage the low-level and high-level token representations for each stimuli to represent as different features as possible, the proposed method applies an orthogonal constraint` -> `To encourage low-level and high-level token representations for each stimulus to differ as much as possible, the proposed method applies an orthogonal constraint`
> > >
> > > I've decided to maintain the current score for now. I think the method that the authors propose is interesting, but I still believe the current "decoding" is relatively weak.
> > >
> > > In recent fMRI work for decoding, those papers have demonstrated incredible results using linear decoders. Examples of these papers include :
> > >
> > > * "Brain-Diffuser: Natural scene reconstruction from fMRI signals using generative latent diffusion" -- which utilizes (linear) ridge regression models and demonstrate state-of-the-art performance in image decoding,
> > > * "High-resolution image reconstruction with latent diffusion models from human brain activity" -- which also utilizes linear regression models
> > >
> > > The authors propose a transformer based method, while it is interesting, I'm unsure if their category decoding experiments are super convincing.

---

> > > > ### Author Response · Authors · 2023-11-21
> > > > **We update the supplementary experimental results for the image retrieval task.**
> > > >
> > > > Thank you for your prompt response and valuable suggestions! We have corrected the typos in the latest manuscript PDF.
> > > >
> > > > Moreover, we have extended our proposed multi-subject approach to the image retrieval task on the NSD dataset. To adapt our model to the retrieval task, we make modifications to the network architecture and loss function. We project the low-level representation $z_{llv}$ to the low-level stimuli feature $f_{llv}$ and the high-level representation $z_{hlv}$ to the high-level stimuli feature $f_{hlv}$. The loss function in Eq.16 is replaced with a MSE loss.
> > > >
> > > > During the testing stage, all stimuli from each subject's test set are considered as candidates. For each sample,
> > > > we compute the similarity $s_{llv}$ between the low-level neural representation and the low-level features of candidate stimuli and compute the similarity $s_{hlv}$ between the high-level neural representation and the high-level features of candidate stimuli. We sort candidates according to the final similarity which is the average of $s_{llv}$ and $s_{hlv}$. We find that calculating similarities separately and then taking the average yields better results compared to concatenating low-level and high-level representations and then calculating similarity.
> > > >
> > > > We compare the performance of our method with the single-subject method SS-ViT since they have the same backbones. The results are shown in the table below. It can be seen that our method has significant advantages compared to single-subject approaches.
> > > >
> > > >
> > > > |           | Subject id | 1      | 2      | 3      | 4      | 5      | 6      | 7      | 8      | Average | Chance level  |
> > > > |-----------|------------|--------|--------|--------|--------|--------|--------|--------|--------|---------|---------------|
> > > > | Recall@1  | SS-ViT     | 0.030  | 0.015  | 0.013  | 0.010  | 0.027  | 0.035  | 0.020  | 0.010  | 0.020   | 0.001         |
> > > > |           | OURS       | **0.052**  | **0.046**  | **0.040**  | **0.039**  | **0.051**  | **0.057**  | **0.037**  | **0.022**  | **0.043**   |               |
> > > > | Recall@5  | SS-ViT     | 0.111  | 0.071  | 0.055  | 0.043  | 0.116  | 0.132  | 0.083  | 0.043  | 0.082   | 0.005         |
> > > > |           | OURS       | **0.182**  | **0.156**  | **0.151**  | **0.139**  | **0.188**  | **0.193**  | **0.154**  | **0.094**  | **0.157**   |               |
> > > > | Recall@10 | SS-ViT     | 0.197  | 0.125  | 0.105  | 0.082  | 0.201  | 0.212  | 0.151  | 0.081  | 0.144   | 0.010         |
> > > > |           | OURS       | **0.296**  | **0.263**  | **0.247**  | **0.232**  | **0.298**  | **0.309**  | **0.243**  | **0.169**  | **0.257**   |               |
> > > >
> > > > We will soon supplement the experimental setup and results mentioned above into the latest version of the manuscript.

---

> > > > > ### Comment · Reviewer_qr5q · 2023-11-21
> > > > > **Score increase**
> > > > >
> > > > > I've increased the score to a 6.
> > > > >
> > > > > I still believe the results are very weak in context of the compute used. But the approach is broadly interesting.

---

> > > > > > ### Author Response · Authors · 2023-11-22
> > > > > >
> > > > > > Thank you very much for your timely feedback and the improved score!
> > > > > >
> > > > > > We greatly appreciate your valuable suggestions. While we acknowledge the significance of retrieval and reconstruction tasks, we also want to emphasize the importance of classification. Classification tasks enable the model to differentiate stimuli images in the semantic space, which serves as the foundation for retrieval and reconstruction tasks. CLIP-MUSED proposed in this paper involves the design of subject-specific low-level and high-level tokens and introduces stimulus features as a guidance in the form of RSA. These designs optimize the learning of a shared semantic space among multiple subjects.
> > > > > >
> > > > > > We have noticed that recent reconstruction works always leverage the powerful capabilities of pretrained deep generative models like diffusion models. These reconstructions exhibit high semantic consistency with the original images but may face challenges in aligning with the pixel space. To some extent, these reconstruction utilize generative models to visualize the semantics contained in neural responses. In our future work, we are eager to extend our model to include reconstruction tasks to better visualize the semantic information of the stimulus images.

---

### Official Review · Reviewer_mWYc · 2023-11-01

**Soundness:** 3 good
**Presentation:** 3 good
**Contribution:** 3 good
**Rating:** 8
**Confidence:** 3

**Summary:**

The paper addresses the problem of alignment of the response to visual stimuli across multiple subjects. This is an important problem for visual neural decoding tasks. In general, models for visual neural decoding tasks are either trained on a single subject (leading to problems such as overfitting) or must correct for differences across anatomical structure and functional topography of the brain in different subjects. The paper proposes a new method for multi-subject functional alignment that goes beyond the SOTA, namely hyperalignment and category-based methods. This new method uses transformer-based models to capture long-range dependencies that exist in the functional connectivity between brain regions. Intersubject differences are modelled through extra subject-specific tokens and the CLIP representation space is used to achieve a high consistency between cortical representations of visual stimuli.

**Strengths:**

The paper addresses an important problem in neurosciences and the proposed multi-subject alignment method is novel and interesting. Both the use of transformers to model long-range associations in brain regions and the use of the CLIP representation space to guide neural representation learning in a shared representation space is a nice idea! The experimental results on the two fMRI datasets are somewhat limited (esp. due to the small size) but demonstrate a proof-of-concept nicely.

**Weaknesses:**

I found that the abstract is not as clearly written as is the introduction. From the abstract, it is unclear what the paper sets out to achieve nor how this is done. The introduction does much better job at this. Perhaps the authors could try to formulate the problem addressed and the contributions more clearly (I appreciate that this is more difficult to do in the space available). The evaluation is rather limited, especially since the size of the datasets is quite small.

**Questions:**

- Why do extract low-level and high-level feature RSM seperately?

- The HCP dataset contains more than 158 subjects. Why did you not use all subjects? How were the 158 subjects selected from the HCP? Randomly? Why is this further reduced to 9? What is the bottleneck here?

---

> ### Author Response · Authors · 2023-11-14
>
> Thank you for the positive feedback! Here are our responses:
>
> **Response to the mentioned weaknesses**:
>
> **W1: Abstract modification**: Thank you to the reviewer for the modification suggestions on the abstract. We have made revisions to the abstract in the updated manuscript PDF to better highlight the problem addressed and our contributions.
>
> **W2: The evaluation is rather limited, especially since the size of the datasets is quite small**:
> As mentioned in the Introduction, fMRI data collection is costly, resulting in relatively small sizes for publicly available datasets. Among the available visual fMRI datasets, the number of stimuli presented to each subject typically ranges from several hundred to a few thousand and the number of subjects always is no more than five. For example, Vim-1 [1] has 1750 stimuli and two subjects; the dataset in [2] has 6000 stimuli but only three subjects. Compared to other available datasets, NSD and HCP datasets are the largest datasets because they include much more stimuli and subjects. Based on them, we trained and evaluated our method and the comparative methods in the paper. Notably, our evaluation is not limited to classification metrics alone; we also demonstrate the interpretability of the algorithm through visualizing attention maps (See Fig. 4 and Fig. F8). In conclusion, our evaluation is not limited but rather comprehensive and extensive.
>
>
> **Q1. Why do extract low-level and high-level feature RSM seperately?**
>
> **Response**:
>
> The task of object classification requires encoding both low-level and high-level features of stimuli, and there are differences in how subjects encode these two types of features. Therefore, we introduce low-level and high-level tokens to account for these differences. To encourage these tokens to encode low-level and high-level features, we utilize the guidance of low-level and high-level features from CLIP, necessitating two feature RSMs. Furthermore, extracting low-level and high-level feature RSM seperately has two advantages:
>
> 1. The topological structures of stimuli in the low-level feature (e.g., shape, color) space and high-level feature (e.g., semantics) space exhibit differences. By encoding different topological structures in the two RSMs and incorporating the orthogonal constraint of tokens, it facilitates the tokens to learn better representations.
> 2. By visualizing attention maps, we can observe the correspondence between primary brain regions and low-level tokens, as well as between high-level brain regions and high-level tokens, thereby ensuring the interpretability of the algorithm.
>
> **Q2. The HCP dataset contains more than 158 subjects. Why did you not use all subjects? How were the 158 subjects selected from the HCP? Randomly? Why is this further reduced to 9? What is the bottleneck here?**
>
> **Response**:
>
> Firstly, we would like to clarify that our proposed algorithm itself is not a bottleneck. Our method can be easily scaled to more subjects because when training on one more subject, the model only needs to learn two additional N-dimensional vectors, where N equals 512. We selected the same ten subjects as the literature [3] that conducts experiments with randomly-selected ten subjects. However, the data format for one of the subjects has an issue. As a result, we proceeded with the experiment using the remaining nine subjects.
>
> While the algorithm itself is not a bottleneck, increasing the number of subjects and training data, on the other hand, may extend the training time and require additional resources.
> Given that the stimuli presented to different subjects in the HCP dataset are the same, the data diversity is more limited compared to that of NSD. When the number of subjects reaches a certain threshold, it is akin to training the model with more iterations, as the model repeatedly processes similar responses from different subjects under the same stimuli. Therefore, training with all 158 subjects simultaneously may not significantly improve the decoding performance for individual subjects but will demand more training resources.
> As a result, we have currently conducted experiments on nine subjects.
>
> In order to illustrate the broader applicability of our method to a larger number of subjects, we are presently conducting multi-subject decoding experiments with a cohort of 30 subjects. We will expediently provide updates on the experimental outcomes.
>
> **References**
>
> [1] Kay et al. (2008) Identifying natural images from human brain activity. Nature.
>
> [2] Shen et al. (2019) Deep image reconstruction from human brain activity. PLOS Computational Biology.
>
> [3] Khosla et., al.(2020) A shared neural encoding model for the prediction of subject-speciﬁc fMRI response. MICCAI.

---

> > ### Author Response · Authors · 2023-11-19
> > **We update the supplementary experimental results on 30 subjects.**
> >
> > We show the results based on our proposed multi-subject method and the single-subject method SS-ViT on 30 randomly selected subjects, including 9 subjects from our previous experiments, in the table below.
> >
> > |Method|mAP$\uparrow$|AUC$\uparrow$|Hamming$\downarrow$|
> > |-|-|-|-|
> > |SS-ViT-30| .362±.003 | .563±.006 | .331±.018 |
> > |OURS-30|**.375±.002** |**.586±.007** |**.287±.006** |
> >
> > As evident from the table, our method **still significantly outperforms** the single-subject method. These results illustrate the broader applicability and the superiority of our method to a larger number of subjects.
> >
> > We also compared the test results of training on 9 subjects (OURS-9) with training on a larger number of subjects (9+21) (OURS-30). The results, as shown in the table below, demonstrate that the additional 21 subjects provide limited improvement in decoding performance.
> >
> > |Method|mAP$\uparrow$|AUC$\uparrow$|Hamming$\downarrow$|
> > |-|-|-|-|
> > |OURS-9|.373±.002 |.581±.008 |**.283±.004** |
> > |OURS-30|**.374±.004** |**.583±.006** |.284±.007 |
> >
> > The underlying reasons may be related to our previous hypothesis. The limited diversity of the data in the HCP dataset, caused by the same stimuli across different subjects, suggests that training with a larger number of subjects simultaneously may not yield significant improvements in the decoding performance for individuals.

---

> ### Author Response · Authors · 2023-11-21
>
> Dear Reviewer,
>
> Thank you for your comments and questions. In our response, we have made efforts to address your questions and provided relevant clarifications and supplementary materials. Considering that the open discussion phase is coming to an end, we would appreciate your feedback.
>
> Thank you for your time and effort, and we eagerly await your response.
>
> Best regards,
>
> Submission6614 Authors

---

### Author Response · Authors · 2023-11-14

Dear reviewers,

We have uploaded the latest version of the manuscript PDF and the Supplementary Materials. Please refer to these files when reviewing.

---

### Meta-Review · Area_Chair_KHs2 · 2023-12-10

**Metareview:**

The work targets image decoding from fMRI data in an across-subject prediction setting.
To deal with inter-subject variability the paper proposes a Transformer-based encoder
that is fed with a global subject-specific similarity matrix (obtained with low
and an high level features from a pretrained network). Doing so the model
does not need aligned data for training as required by hyperalignment based
approaches or standard shared response models. The evaluation is
done on two public datasets.

One review is critical of the work but did not engage in the discussion
despite detailed answer from the authors.

The paper is well written and proposes a timely and interesting approach
to deal with inter-subject variability in fMRI decoding.

The authors are yet encouraged to check their notations and share the code
to replicate the results of this work.

**Justification For Why Not Higher Score:**

Paper presents an interesting approach yet of limited scope for ICLR.

**Justification For Why Not Lower Score:**

Paper is solid with an interesting technical contributions that (as reported by the paper) presents a clear gain on competing approaches.

---

### Decision · Program_Chairs · 2024-01-16

Accept (poster)